# Environmental sustainability assessment of biodiesel production from *Jatropha curcas* L. seeds oil in Pakistan

Taslima Khanam[1], Faisal Khalid[2], Wajiha Manzoor[3], Ahmad Rashedi🄶[1], Rana Hadi[4], Faizan Ullah[5], Fariha Rehman[3], Andleeb Akhtar[6], N. B. Karthik Babu[7], Majid Hussain🄶[2]*

1 College of Engineering, IT and Environment, Charles Darwin University, Casuarina, Northern Territory, Australia, 2 Department of Forestry and Wildlife Management, University of Haripur, Haripur City, KP, Pakistan, 3 Department of Economics, COMSATS University Islamabad (CUI), Lahore Campus, Lahore City, Punjab, Pakistan, 4 Department of Zoology, Jinnah University for Women, Nazimabad, Karachi, Sindh, Pakistan, 5 Department of Botany, University of Science and Technology, Bannu, KP, Pakistan, 6 Department of Psychology, University of Haripur, Haripur City, KP, Pakistan, 7 Department of Mechanical Engineering, Centurion University of Technology and Management, Bhubaneswar, Odisha, India

* majid@uoh.edu.pk

**Data Availability Statement:** All relevant data are within the manuscript and its Supporting Information files.

## Abstract

According to IPCC Annual Report (AR-5), environmental impact assessment of any product prototype is recommended before its large-scale commercialization; however, no environmental profile analysis of any biodiesel prototype has been conducted in Pakistan. Therefore, objective of this study was to conduct a comprehensive life cycle assessment (LCA), water footprint and cumulative energy demand (CED) of biodiesel production from *Jatropha curcas L.* (JC) seeds oil in Pakistan. A cradle-to-gate LCA approach was applied for 400 liter (L) *JC* biodiesel produced in Pakistan. *JC* biodiesel production chain was divided into three stages i.e., 1). cultivation of *JC* crop 2). crude oil extraction from *JC* seeds and 3). crude oil conversion to biodiesel. Primary data for all the stages were acquired through questionnaire surveys, field visits and measurements in the field. Potential environmental impacts were calculated in SimaPro v.9.2 software using Eco-indicator 99 methodology. Results showed that crude oil extraction stage accounted for highest emissions (77%) to the overall environmental impact categories evaluated, followed by oil conversion stage (21%) and *JC* cultivation stage (02%), respectively. The three stages of *JC* biodiesel production chain are major contributor to ecotoxicity with a contribution of 57% to this impact category. Higher contribution to ecotoxicity was due to agrochemicals used in the *JC* cultivation. Similarly, fossil fuels impact category was responsible for 38% of overall environmental impacts. In addition, water footprint of *JC* biodiesel production chain was 2632.54 $m^3$/reference unit. Cumulative energy required for 400L *JC* biodiesel production chain was 46745.70 MJ in Pakistan. Fossil diesel consumption, synthetic fertilizers use and purchased electricity were major hotspot sources to environmental burdens caused by *JC* biodiesel production in Pakistan. By performing sensitivity analysis at 20% reduction of the baseline values of fossil diesel used, synthetic fertilizers and purchased electricity, a marked decrease in environmental footprint was observed. It is highly recommended that use of renewable energy instead of fossil energy would provide environmental benefits such as lower greenhouse gases and other

**Funding:** The author(s) received no specific funding for this work.

**Competing interests:** The authors have declared that no competing interests exist.

toxic emissions as compared to conventional petroleum fuels. It is also recommended that *JC* as a biofuel plant, has been reported to have many desired characteristics such as quick growth, easy cultivation, drought resistance, pest and insect resistance, and mainly great oil content in JC seeds (27–40%). Therefore, *JC* plant is highly recommended to Billion Tree Afforestation Project (BTAP) for plantation on wasteland because it has multipurpose benefits.

## Introduction

Globally, energy utilization has increased due to improved living standards and growing population, particularly after the start of the 20[th] century industrial revolution [1]. A further 53 percent rise in global energy consumption by 2030 is expected by the International Energy Agency [2]. Currently, this energy comes from fossil fuels sources such as crude oil (33%), coal (30%) and natural gas (24%) [3–5]. In the transport sector, almost all energy comes from the crude oil (98%) [6] and energy consumption in that sector is predicted to rise by an average of 1.8% per annum from 2005 to 2035 [7, 8]. High consumption levels of non-renewable fossil fuels are resulting climate change and threaten the energy security of people with less access to these resources. This situation is motivating a search for alternatives to fossil fuels as energy sources [9] in order to mitigate climate change and to ensure energy security [10]. The aim is to develop and find a renewable, sustainable and economically feasible alternative sources of energy [11]. Biomass energy such as biodiesel is one potential alternative.

Over the past twenty years, biodiesel produced from different biomass feedstocks has undergone much research and development [12, 13]. *Jatropha curcas (JC)* as a biodiesel feedstock has various desired characteristics such as fast growth, easy cultivation, good insect and pest resistance, and high oil content seeds that are suitable for biodiesel fuel production [14, 15]. In India the scarcity of Jatropha seeds continues to be a major barrier to rising biodiesel production, furthermore, biodiesel projects have failed due to the limitations such as low Jatropha seed production, small availability of wasteland, and high plantation and maintenance costs [16]. Unfortunately, no substantial progress has been made, and Jatropha has not made a significant contribution to the energy scenario due to a lack of high-yielding cultivars, large-scale plantation without the evaluation of planting materials, knowledge gap and basic research gap [17]. Large-scale Jatropha plantation is hampered by a lack of research on planting techniques and inadequate management of the planting base [18, 19]. *JC* biodiesel has the ability to offer a potentially energy-rich and commercially feasible alternative to fossil fuels, as it has similar physio-chemical characteristics to fossil diesel [20]. *JC* is a drought tolerance plant that needs few nutrients and little management [21]. *JC* plants grown along the agriculture fields can act as shelterbelt by protecting the agricultural land from soil erosion [22, 23]. Generally, *JC* plant reaches to maturity in 6 years and production of seeds continues for the next 30–40 years [24]. The yield of *JC* seed in ranges up to 15 ton/hectare/year [25] and the crude oil content of *JC* seed ranges from 30–35 percent [26]. Thus, the yield of crude oil from *JC* seed has been reported to range from 158–396 gallons per hectare [27]. Around 980 g of pure biodiesel can be produced from 1000 g of *JC* crude oil [28]. The primary raw materials for JC cultivation included polythene bags, *JC* cuttings, insecticide, pesticide, synthetic fertilizers, green manure, and diesel fuel consumed in tractor for transportation of the seeds. Materials for oil extraction included the *JC* seeds, methanol, sodium hydroxide, water and fossil diesel, and materials for oil conversion were *JC* seed crude oil, methanol, sodium hydroxide, steam and water. By-

products of *JC* biodiesel production chain are glycerine and press cake. Glycerine can be used in medicines for skin care, cosmetics and soap making, whereas press cake can be used as a food for fishes and organic fertilizer such as bio-compost [29, 30].

Pakistan is fossil fuels importing country and 17.20 million tons of crude oil were imported in the fiscal year 2018–2019 [31, 32]. In Pakistan transport and electricity sectors are the key user of fossil fuel and would need about 50 percent extra energy for electricity and transport sector in near future [33]. Main energy consumption sector of Pakistan includes residential, private, agriculture, manufacturing, forestry, transportation and other government sectors [34]. Pakistan desires to blend 10 percent biodiesel in fossil diesel by 2025 [35]. Thus, research work on the lab-scale has been conducted on biodiesel production in numerous universities and organizations in the country [36–40]. Over the past twenty year's biodiesel has undergone several research and improvement studies [12, 13, 41]. According to Chakrabarti *et al.*, 2012 many organisations have been working on production of biodiesel prototypes development from different biomass sources in Pakistan. So far, biodiesel is produced as a prototype in Pakistan which proves to be a good source of renewable and sustainable energy. The Quaid-e-Awam University (QUEST) at Nawabshah [42], The University of Engineering and Technology (UET) Lahore [43], The Institute of Chemistry at Punjab University [44], The Quaid-e-Azam University Islamabad [45], NUST Islamabad [46], NED University of Engineering and Technology, Karachi [47], The University of Agriculture, Faisalabad [44], and University of Science and Technology, Bannu, Khyber Pakhtunkhwa has prepared biodiesel prototypes from *Jatropha curcas* seeds oil in Pakistan. However, there has been no environmental analysis of any of the biodiesel prototypes developed in Pakistan. Therefore, the objective of this study was to conduct a comprehensive life cycle assessment (LCA) of a biodiesel prototype produced from *Jatropha curcas* seeds oil in Pakistan, which included calculation of material and energy flows, water footprint, cumulative energy demand (CED) and emissions to soil, water and air. Moreover, sensitivity analysis was also conducted to evaluate the potential impacts of *JC* biodiesel production if applied on a large-scale in Pakistan.

## Materials and methods

### Study design

Life Cycle Assessment (LCA) is the estimation of inputs, outputs and potential environmental impacts of a product system during its entire life cycle stages [12, 13, 41, 42–47]. Typically, the LCA involves: (a) goal and scope definition (b) life cycle inventory (LCI), (c) life cycle impact assessment (LCIA) and (d) Results interpretation [44].

### System boundary and reference unit

The system boundary of the present study is presented in (Fig 1). The biodiesel production chain was divided into three stages i.e., 1). Cultivation, 2). Oil extraction and 3). Oil conversion to biodiesel. A cradle-to-gate LCA approach was followed in the present study. The reference unit was defined as 400 litres of biodiesel produced from *JC* seeds at the production facility in the Biodiesel Laboratory of UST, Bannu, Pakistan.

### Life cycle inventory and impact assessment

A detailed questionnaire regarding the inputs and outputs of biodiesel production chain was sent to the Biodiesel Laboratory of the Department of Botany, University of Science and Technology (UST), Bannu, KP, Pakistan. The primary data were collected for cultivation, lab-based

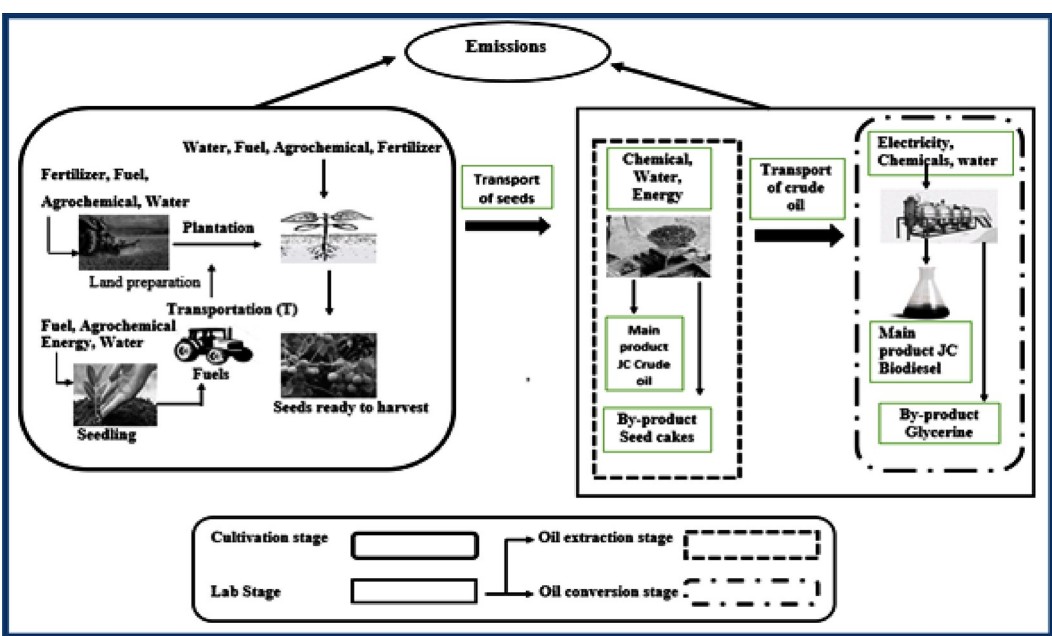

**Fig 1. System boundary of the study.**

biodiesel production (crude oil extraction stage and oil conversion to biodiesel stage) as presented in (Table 1). The secondary data were taken from published literature and Ecoinvent database. The data were modelled using SimaPro 9.0 software and applying Eco-indicator 99E methodology. Sensitivity analysis was conducted as per ISO 2006 protocol 14040–14043 to examine the impact of varying an input parameter such as diesel fuel consumed in transport of materials, synthetic fertilizers applied in the *JC* nursery and cultivation stage and purchased electricity consumed in machinery at the oil conversion stage.

## Mass and economic allocation and assumptions

If a process generates more than one valuable output, allocation of environmental impacts to each output is essential [45]. In the present study, both mass and economic based allocation were applied. In the oil extraction stage of biodiesel production from *JC* seeds, the mass-based allocation was applied for the main product (crude oil, 58%) and the co-product (seedcake, 42%), whereas in the oil conversion stage, economic allocation was applied for the main product (biodiesel oil) and co-product (glycerine). JC biodiesel was accounted for 95% of environmental burdens (0.957 US $/L biodiesel in 2019) and purified glycerine was for 5% (0.209 US $/L in 2019) based on economic allocation.

Further assumptions for this study included;

a. Water consumption data for the *JC* plants cultivation, oil extraction and oil conversion stage were taken from Onabanjo et al. 2015 and Rodríguez et al. 2018 [48].

b. Heat required during the oil conversion stage was converted to joules (J) and mega joules (MJ) using the formula ($Q = c^*m^*\Delta T$) Where, Q is the quantity of heat, c is specific heat of the substance or solution and its unit is joules per gram Celsius, m is the mass of the substance or solution in kg (kilogram) and $\Delta T$ is change in Temperature and its unit is kelvin.

**Table 1. Life cycle inventory of inputs/outputs to produce 400L biodiesel from *JC* seeds oil in Pakistan.**

| Cultivation stage | | |
|---|---|---|
| **Resources input** | **Unit** | **Value/ quantity** |
| Cultivated land | Square meter | 505.857 |
| Space between plants to plant | m (meter) | 1.8 |
| Space between row to row | m | 1.8 |
| Seedling density | No of plants /kanal | 600 |
| Seeds/cuttings used for rising nursery | Number | 500 |
| Polythene bags | Number | 650 |
| Electricity is used in *JC* nursery | kWh/day | 10 |
| Insecticide (Karate) used in nursery | kg/kanal | 0.08 |
| Insecticide (Curocron) used in nursery | kg/kanal | 0.16 |
| Urea | kg/kanal | 2.507 |
| DAP (Di-ammonium phosphate) | kg/kanal | 3.091 |
| Potassium (K) | kg/kanal | 1.002 |
| Green Manure or organic compost | kg/kanal | 0.5 |
| Fungicide is used in nursery (Ridomil) | Kg/kanal | 5E-02 |
| water is consumed | L/year.kanal | 6836 |
| Transportation distance of seedling to plantation site | tkm | 0.096 |
| Diesel Fuel (HSD) consumed by Vehicles (tractor) for transportation of seedling to field | L | 18 |
| Time of irrigation in summer | per Month | 3 |
| Time of irrigation in winter | per Month | 2 |
| **Output (Seed produced from 1 kanal cultivation of *JC* in Pakistan** | **Kg** | **2259** |
| Manufacturing/Lab stage | | |
| Oil Extraction stage | | |
| **Resources input** | **Unit** | **Value** |
| Quantity of seed used for extraction of oil | kg | 2259 |
| Delivery distance of seed from field to extraction site | tkm | 0.058 |
| Chemical used (Methanol) | kg | 80 |
| Sodium hydroxide (NaOH) | kg | 4.8 |
| Water | L | 50 |
| **Output oil extracted** | **L** | **565** |
| **By-product (Seed cake) (residue press)** | **kg** | **1882** |
| Oil conversion stage | | |
| **Resources input** | **Unit** | **Value** |
| Crude Oil extracted from *JC* seeds | L | 565 |
| Electricity used for oil conversion | kWh | 1000 |
| Chemicals used (Methanol) | kg/L | 80 |
| Sodium Hydroxide NaOH | kg/L | 4.8 |
| Steam heat is required | MJ | 37.6 |
| Water | L/400L biodiesel | 200 |
| **Output Quantity of biodiesel produced** | **L** | **400** |
| **By-product (purified Glycerine)** | **L** | **94** |

c. Soya bean oil process data present in Ecoinvent database was used instead of *JC* crude oil in oil conversion stage, because the *JC* crude oil was not available in the SimaPro v 9.2 software.

### Estimation of water footprint of JC biodiesel production chain

The data for the *JC* cultivation stage, i.e., crop phenology, irrigation, and field management, were collected from field experiments that took place at the University of Science and Technology, Bannu, Pakistan during 2019–2020. The annual green and blue water footprint for the *JC* cultivation stage was estimated using AquaCrop Model version 6.1 developed by FAO (Food and Agriculture Organization) [46, 47]. According to the Hoekstra et al., 2011 blue water footprint is indicator of freshwater use (surface and groundwater) for person or community development of goods and services and green water footprint is utilization of rainwater, which does not run off, or refill the groundwater but is retained as soil moisture within the soil. Then, the green and blue water footprint of *JC* plantation for Bannu was summed up with the blue water footprint of Lab-based *JC* biodiesel production stage. For Lab-based *JC* biodiesel production stage, blue water footprint was estimated using Hoekstra et al. 2011 method present by-default in SimaPro version 9.0 software. The blue and green water footprint of JC crop was estimated following the global water footprint accounting standards [44]. The AquaCrop model of FAO (version 6.1) was used to simulate the soil water balance and *JC* productivity [49, 50]. This model estimates the evapotranspiration (ET) and *JC* yield by simulating the dynamic soil water balance and biomass growth on a daily basis (Eq 1).

$$S_i = R_i + I_i + CR - RO_i - Dr - ET_i \qquad \text{Eq (1)}$$

Where; S is soil water content (mm) on day *i*, R is rainfall (mm), I is mean irrigation (mm), CR is capillary rise (mm), RO is mean surface runoff, Dr is drainage (mm) and ET is evapotranspiration [51]. The output of the AquaCrop simulation—crop growth characteristics and water fluxes were partitioned into blue and green parts using the method introduced by [52]. The blue and green components of *JC* crop water use (CWU) was assessed by summing the blue and green ET over the *JC* crop growing period as shown in Eqs 2 and 3.

$$CWU_b = \sum_{t=1}^{T} \frac{S_{bt}}{S_t} ET_{t \times 10} \qquad \text{Eq (2)}$$

$$CWU_g = \sum_{t=1}^{T} \frac{Sg_t}{S_t} ET_{t \times 10} \qquad \text{Eq (3)}$$

Where, $CWU_b$ and $CWU_g$ are blue and green water consumption (m$^3$), $S_{bt}$ and $S_{gt}$ are change in blue and green soil water stock over the growing season and 10 is the conversion factor from mm to m$^3$. The $WF_b$ and $WF_g$ were obtained by dividing CWU by the crop yield Y using Eqs 4 and 5 [52].

$$WF_b = \frac{CWU_b}{Y} \qquad \text{Eq (4)}$$

$$WF_g = \frac{CWU_g}{Y} \qquad \text{Eq (5)}$$

### Sensitivity analysis for identification of clean and green options in *JC* biodiesel production chain

A sensitivity analysis was conducted as per ISO protocol 2006; 14000 series that involve examining the impact of varying an input parameter. Sensitivity analysis was performed for identification of clean and green options in the *JC* biodiesel production chain using SimaPro v9.1 software. A 20% reduction in the baseline values for fossil diesel use, synthetic fertilizers and purchased electricity was applied to check its effect on the emissions reduction or

improvement in the environmental profile of the *JC* biodiesel production in Pakistan. A sensitivity analysis was carried out to reduce the hotspot sources to environmental impacts caused by *JC* biodiesel production without going to deteriorate the quality of the biodiesel to encourage clean and green production of biodiesel from *JC* seeds in Pakistan.

## Results and discussion

### Results

**Environmental impacts of JC biodiesel production chain.** The results of life cycle impact assessment for 400L *JC* biodiesel production are presented in this section. In this study, the US-EPA top ten most wanted environmental impact categories were evaluated i.e., carcinogens, respiratory organics, respiratory inorganics, climate change, ozone layer depletion, eco toxicity, acidification/ eutrophication, and fossil fuels [53, 54]. The overall results for environmental impact categories of three stages i.e. *JC* cultivation stage, oil extraction and oil conversion stage are presented in Table 2. Among the three stages, the oil extraction stage has major contribution (77%) to all the environmental impact categories, followed by the oil conversion stage (21%), respectively. The cultivation stage has minor contribution (2%) to all the environmental impact categories as can be seen in Fig 2. The highest environmental impact of the three stages of biodiesel production from *JC* seeds was posed by ecotoxicity (3828.784 PAF*m2yr), followed by fossil fuels (2502.347 MJ surplus) and acidification/eutrophication (341.231 PDF*m2yr). The carcinogens (0.0216 DALY), respiratory inorganics (6.6E-03 DALY), climate change (1.2E-03 DALY), respiratory organics (5.2E-06 DALY) and ozone layer depletion (2.3E-07 DALY) accounted for minor environmental impacts. The overall results showed that the three stages of biodiesel production (*JC* cultivation, crude oil extraction and oil conversion) were major contributor to ecotoxicity impact category with a contribution of 57% to the overall impact categories evaluated. Similarly, the overall emissions accounted for 38% of the environmental burdens in the fossil fuels impact category. Moreover, the contribution of cultivation stage, oil extraction stage and oil conversion stage to acidification/eutrophication was 5% to all the environmental impact categories as can be seen in Fig 3. Emissions to different environmental compartments such as air, water and soil from the three stages of JC biodiesel production chain in Pakistan are summarized in the S1–S9 Tables.

**Cumulative energy demand (CED) for JC biodiesel production chain.** The CED results and related hotspots of the three stages of *JC* biodiesel production are presented in Table 3. The total CED required for the three stages were amounted to 46745.70 MJ from energy sources i.e., non-renewable fossil, non-renewable nuclear, non-renewable biomass and renewable water. However, among the energy sources, non-renewable fossil had the highest

**Table 2. Environmental impacts of the three stages of biodiesel production from *JC* seeds oil.**

| Impact category | Unit | Total value | Cultivation stage | Oil extraction stage | Oil conversion stage |
|---|---|---|---|---|---|
| **Carcinogens** | DALY | 0.0216 | 0.0004 | 0.020 | 0.0009 |
| **Respiratory organics** | DALY | 5.2E-06 | 3.4E-08 | 7.7E-07 | 4.4E-06 |
| **Respiratory inorganics** | DALY | 6.6E-03 | 3.0E-05 | 0.003 | 0.004 |
| **Climate change** | DALY | 1.2E-03 | 6.0E-06 | 0.0003 | 0.0008 |
| **Ozone layer** | DALY | 2.3E-07 | 1.4E-08 | 1.2E-07 | 9.4E-08 |
| **Ecotoxicity** | PAF*m2yr | 3828.784 | 6.838 | 3536.916 | 285.030 |
| **Acidification /Eutrophication potential** | PDF*m2yr | 341.231 | 0.669 | 303.960 | 36.602 |
| **Fossil fuels** | MJ surplus | 2502.347 | 103.879 | 1291.808 | 1106.660 |
| **Grand total** | | | **111.3860** | **5132.7085** | **1428.2966** |

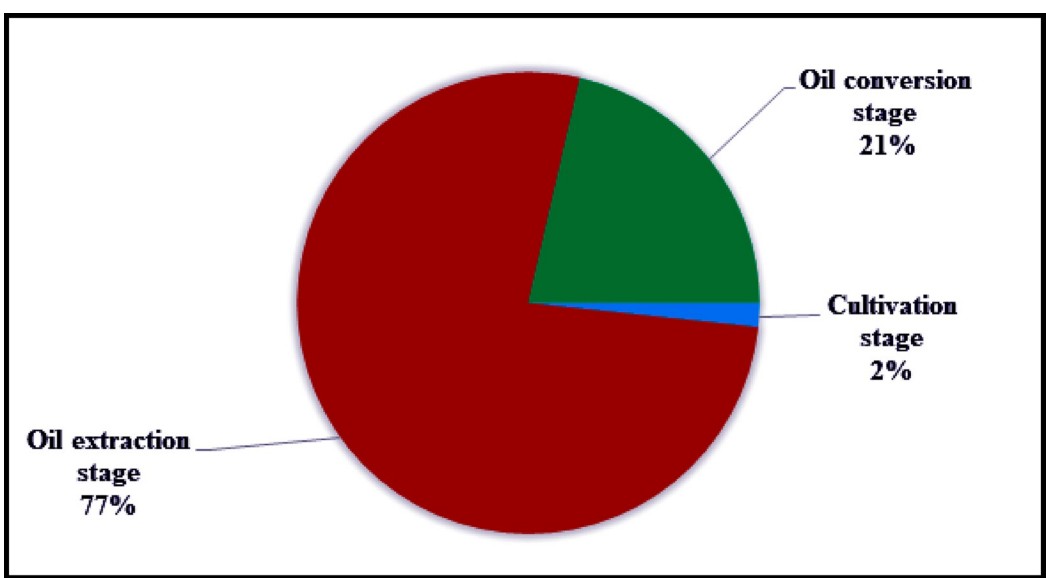

**Fig 2. Percent contribution per process of three stages of JC biodiesel to various environmental impact categories.**

contribution i.e., 32982.40 MJ to the total energy removed from the nature to produce 400 L *JC* biodiesel as a prototype in Pakistan. The major consumption from the non-renewable fossil fuel source was in oil extraction stage of *JC* biodiesel production, followed by oil conversion stage, and cultivation stage, respectively. Whereas, non-renewable biomass was the second largest source with contribution of 9829.50 MJ energy use. Likewise, highest energy was consumed by oil conversion stage from non-renewable biomass, followed by oil extraction stage, and cultivation stage for 400 L biodiesel production from *JC* seeds. The renewable water had the contribution 1212.7 MJ to the total energy used by the process. The conversion stage was the single major contributor to renewable water, followed by extraction stage, and cultivation stage. Concisely, most of the energy was used from non-renewable fossil during the production of biodiesel from *JC* seeds. Maximum energy was consumed by oil conversion stage (59%), followed by oil extraction stage (38%), and cultivation stage (3%) as can be seen (Fig 4).

**Sensitivity analysis for clean and green options in JC biodiesel production chain.** Around 20% reduction of fossil diesel use with respect to its baseline value was assumed which leads to 17.32% decrease in environmental burdens in the ozone layer depletion, 15.57% in respiratory organic, 15.42% in fossil fuels, 8.09% in respiratory inorganic, 6.32% in climate change, 5.99%, 4.21% and 1.80%, in ecotoxicity, carcinogens and acidification potential/eutrophication potential, respectively. Similarly, the use of urea was also assumed to be decrease by 20% from its baseline value, the impacts of urea in acidification potential /eutrophication potential was dropped by 5.17%, carcinogens was decreased up to 4.16%, respiratory inorganic was decreased by 1.53%, 1.2% decrease occurred in climate change, ozone layer depletion was decreased by 1.14%, respiratory organic was decreased by 0.6% and ecotoxicity was decrease up to 0.47% as summarized in (Table 4). Similarly, 20% reduction in the use of DAP, all-environmental impact categories were decreased i.e., acidification potential/eutrophication potential (6.03%) carcinogens (4.16%), respiratory inorganic (2.65%), climate change (1.77%), fossil fuels (1.19%), ozone layer depletion (1.10%), respiratory organic (1.09%) and the reduction in impact to the ecotoxicity was (0.01%). The reduction in potassium fertilizer uses by 20% leads to the maximum percentage of decrease in carcinogens (23%), followed by AP/EP (8.09%),

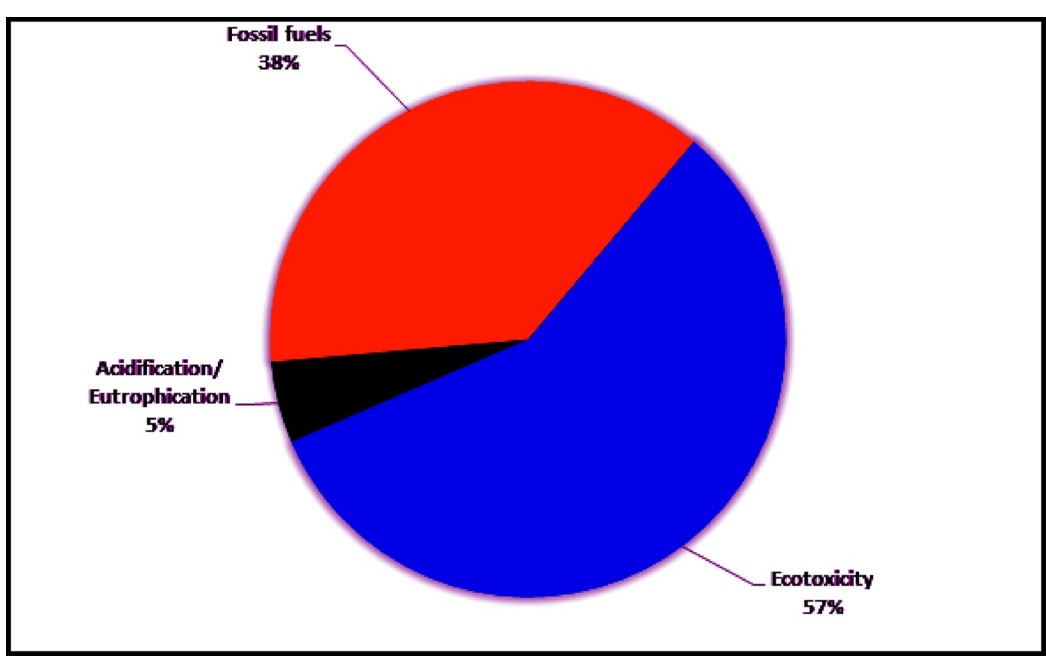

**Fig 3. Percent contribution of biodiesel production chain to environmental impact categories.**

respiratory inorganic (2.62%), ecotoxicity (1.30%), climate change (1.07%), ozone layer depletion (1.01%), respiratory organic (1%) and fossil fuel (0.21%), respectively. In the oil conversion stage, the use of electricity was reduced by 20%, so the climate change impact was decreased by 20.12%, followed by the reduction in the respiratory inorganic environmental impact category (12.59%). With the 20% reduction in the electricity, the 11.22% decrease occurred in carcinogens impact category and ecotoxicity was reduced up to 10.81%, respectively. The reduction occurred in fossil fuels was (9.79%), acidification potential /eutrophication potential was 6.70%, ozone layer depletion (5.34%) and respiratory organic was decrease up to (0.49%) as presented in (Table 4).

**Water footprint (WF) of JC biodiesel production chain.** The total WF for the three stage of *JC* biodiesel production was equal to 2632.54 m³/400L biodiesel production in Pakistan with most of the water consumption occurred in the cultivation stage. Among the three stages of *JC* biodiesel production chain, the total WF for cultivation stage was estimated to 2460 m³/400L biodiesel, whereas the total WF for crude oil extraction stage and oil conversion stage was equal to 154 m³ and 18.54 m³ per 400L biodiesel from *JC* seeds. Maximum water was consumed by cultivation stage, followed by oil extraction stage and then oil conversion stage as can be seen in (Fig 5).

**Table 3. Cumulative energy demand of *JC* biodiesel production chain in Pakistan.**

| Impact category | Non-renewable, fossil | Non-renewable, nuclear | Non-renewable, biomass | Renewable, water |
|---|---|---|---|---|
| Cultivation stage | 1321.3 | 45.8 | 1.9 | 13.8 |
| Oil extraction stage | 16748.9 | 840.3 | 35.1 | 367.9 |
| Oil conversion stage | 14912.2 | 1835.0 | 9792.5 | 831.0 |
| **Total** | **32982.4** | **2721.1** | **9829.5** | **1212.7** |

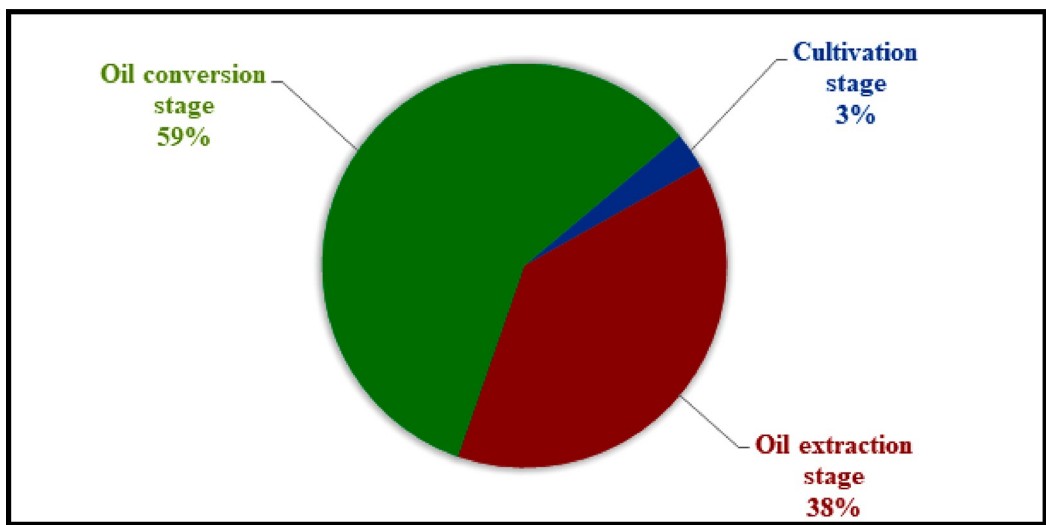

**Fig 4. Percent energy consumption by JC biodiesel production chain.**

## Discussion

Our results are similar with previous research studies, where environmental burdens were primarily associated with fossil diesel use, electricity consumption, and the use of synthetic fertilizer [45, 55–60]. Our findings were in accordance with previous studies conducted on biodiesel production from different biomass sources such as *JC*, soybean, castor oil, waste cooking oil etc, where oil extraction stage accounted for the largest contribution and cultivation stage was responsible for minor emissions to all the environmental impact categories evaluated [61] With respect to GHG emissions, the main impact is related to *JC* seeds processing (79.3%), whereas 20.7% emissions were caused by cultivation of *JC* seeds [62]. A comprehensive LCA of biodiesel production from *JC* in Thailand by [63] also showed that the biodiesel production stage led to higher emissions in all the environmental impact categories relative to cultivation stage of biodiesel production from *JC* seeds oil. The primary reason for this higher contribution of the production or manufacturing stage of biodiesel to all environmental impact categories is due to the use of chemicals such as sodium hydroxide (NaOH) and methanol, and energy consumption during the manufacturing processes [64].

Ecotoxicity impacts were mainly due to synthetic fertilizers, insecticide and pesticide use in the cultivation stage, and chemicals used in the manufacturing stage [65]. The impact of acidification/eutrophication is mostly because of the emissions of ammonia and nitrogen oxide to air from synthetic fertilizers. The primary reason for these higher emissions was from synthetic fertilizers use, diesel fuel consumed in the vehicle for transportation of saplings to the plantation site and preparation of land for cultivation through tractor mechanization [66]. However, in the present study, *JC* nursery and *JC* cultivation field was established within the University of Science and Technology, Bannu, Pakistan, where less distances were covered to transport *JC* saplings from nursery to cultivation site, *JC* seeds from field to biodiesel production site/Lab, thus consuming less quantities of fossil fuels and other resources.

Cumulative energy demand refers to the amount of energy extracted from nature to produce a specific product such as biodiesel [67]. Maximum energy was consumed by oil conversion stage (59%), followed by oil extraction stage (38%), and the cultivation stage (3%) in this study. Our results were lower than the [68], where cultivation stage was responsible for 45.7% emissions, followed by oil extraction stage (15.7%) and oil conversion stage (38.5%) stage. The

**Table 4. Comparative environmental impacts assessment of baseline results with the results obtained by 20% reduction in the following inputs consumption.**

| Hotspot sources | Impact category | Carcinogens | Respiratory organic | Respiratory inorganic | Climate change | Ozone layer depletion | Ecotoxicity | Acidification potential /eutrophication potential | Fossil fuels |
|---|---|---|---|---|---|---|---|---|---|
| | Unit | DALY | DALY | DALY | DALY | DALY | PAF*m2yr | PDF*m2yr | MJ surplus |
| **Diesel use** | Baseline value of diesel use (18 L) | 0.0004 | 3.4E-08 | 3.0E-05 | 6.0E-06 | 1.4E-08 | 6.84 | 0.7 | 103.88 |
| | 20% reduction in diesel use (14.4 L) | 0.0004 | 2.9E-08 | 2.8E-05 | 5.6E-06 | 1.2E-08 | 6.39 | 0.62 | 87.88 |
| | Percent decrease in environmental impacts | 4.21% | 15.57% | 8.09% | 6.32% | 17.32% | 5.99% | 1.80% | 15.42% |
| **Urea** | Base line value of urea (2.507 kg) | 0.0004 | 3.4E-08 | 3.0E-05 | 6.0E-06 | 1.4E-08 | 6.8 | 0.7 | 103.9 |
| | 20% reduction in urea use (1.64 kg) | 0.0004 | 3.4E-08 | 3.0E-05 | 5.9E-06 | 1.4E-08 | 6.83 | 0.66 | 102.01 |
| | Percent decrease in environmental impacts | 4.16% | 0.67% | 1.53% | 1.24% | 1.13786 | 0.47748 | 5.17% | 1.82% |
| **DAP** | Base line value of DAP use (3.091 kg) | 0.0004 | 3.4E-08 | 3.0E-05 | 6.0E-06 | 1.4E-08 | 6.8 | 0.7 | 103.9 |
| | 20% reduction in DAP use (2.47 kg) | 0.0004 | 3.4E-08 | 2.9E-05 | 5.9E-06 | 1.4E-08 | 6.80 | 0.66 | 102.66 |
| | Percent decrease in environmental impacts | 4.16% | 1.09% | 2.65% | 1.77% | 1.09571 | 0.01% | 6.03% | 1.19% |
| **Potassium** | Base line value of Potassium use (1.002) | 0.0004 | 3.4E-08 | 3.0E-05 | 6.0E-06 | 1.4E-08 | 6.8 | 0.7 | 103.9 |
| | 20% reduction in Potassium use (0.802 kg) | 0.0003 | 3.4E-08 | 2.9E-05 | 5.9E-06 | 1.4E-08 | 6.71 | 0.64 | 103.68 |
| | Percent decrease in environmental impacts | 23.00% | 1.00971% | 2.62% | 1.07% | 0.01 | 1.30% | 8.09% | 0.21% |
| **Electricity** | Base line value of electricity use (1000 kWh) | 0.001 | 4.4109E-06 | 0.004 | 0.001 | 9.40558E-08 | 285.03 | 36.60 | 1106.7 |
| | 20% reduction in electricity use (800 kWh) | 0.0009 | 4.4E-06 | 0.003 | 0.0008 | 8.9E-08 | 254.22 | 34.15 | 998.33 |
| | Percent decrease in environmental impacts | 11.22% | 0.49% | 12.59% | 20.12% | 5.34% | 10.81% | 6.70% | 9.79% |

primary reason for lower emissions in the cultivation stage was attributed to the fact that *JC* nursery and cultivation was established on small-scale on campus site with minimum resource inputs and mechanization as compared to [69]. Furthermore, in the oil conversion stage in *JC* biodiesel production in Pakistan, mostly purchased electricity and fossil energy was consumed to run equipment/machinery for crude *JC* oil conversion into *JC* biodiesel. Energy demand for the processing stage of biodiesel alone accounted for 65% and 85% of the overall energy demand for palm and JC biodiesel manufacturing, respectively [56–65]. Likewise, according to, *JC* cultivation stage accounted for 12% to the overall CED, whereas overall transportation of *JC* saplings to the plantation site, oil cake and unrefined Jatropha oil contributed 15% to the CED in China. Our results were less than [66], because our analysis was based on Lab-based pilot study, where minimal distances were covered and little quantities of chemicals were

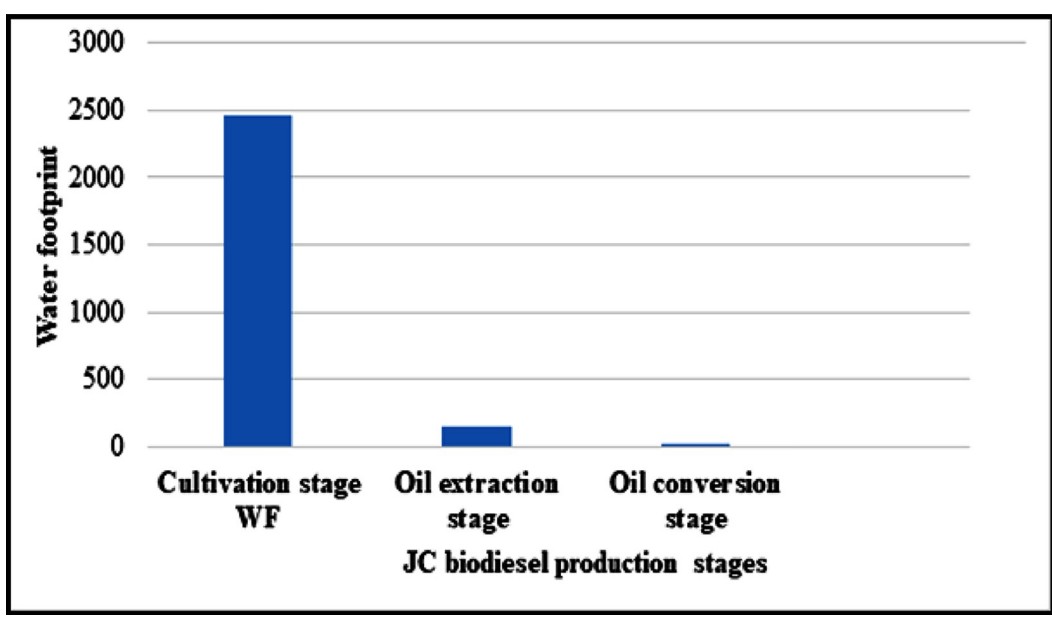

**Fig 5. Water footprint of JC biodiesel production chain in Pakistan.**

utilized during the entire production chain of the *JC* biodiesel. However, our results were in line with Ndong et al. 2009 for manufacturing stage of *JC* biodiesel production, where transesterification consumed huge quantity of purchased electricity (61%) [67]. Similarly, [65–68] reported that the *JC* oil extraction and processing has higher consumption of energy as compare to cultivation stage, which is in accordance with the results of the present study [62]. reported that water footprint is mostly due to cultivation stage in the *JC* biodiesel production chain, thus irrigation is one of the most significant features for the development of this biofuel plant. In the cultivation stage, water use is higher relative to the manufacturing/Lab stage [69, 70] which is similar to our results in the present study.

## Conclusions and recommendations

This is the ever first study on the environmental sustainability, water footprint and cumulative energy demand of *Jatropha curcas* (*JC*) biodiesel production chain in Pakistan. From the results, it is concluded that most of the hotspot's inputs were from the cultivation stage of the biodiesel production chain such as fossil diesel and synthetic fertilizers like urea, DAP, potassium, whereas from oil conversion stage, purchased electricity consumption was the major contributor to overall environmental footprints. By performing sensitivity analysis at 20% reduction in the baseline values for fossil diesel use, synthetic fertilizers and purchased electricity, a marked decrease in most of the environmental impact categories were observed. The present study provides a benchmark or baseline for future research work to investigate material and energy flows, environmental burdens, water footprint of biofuels production from other bioenergy crops in Pakistan. Besides biodiesel production from *JC* seeds oil, its by-products can also be used as a value-added raw material in pharmaceutical, soap and bio pesticides manufacture industries.

## Supporting information

**S1 Table. Emissions to water from cultivation of *JC* plantation during 2019–2020 in Pakistan.**
(DOCX)

**S2 Table. Emissions to soil from cultivation of *JC* plantation in Pakistan during 2019–2020.**
(DOCX)

**S3 Table. Emissions to air from cultivation of *JC* plantation in Pakistan during 2019–2020.**
(DOCX)

**S4 Table. Emissions to water from JC oil extraction phase in Pakistan during 2019–2020.**
(DOCX)

**S5 Table. Emissions to soil from JC oil extraction phase in Pakistan during 2019–2020.**
(DOCX)

**S6 Table. Emissions to air from JC oil extraction phase in Pakistan during 2019–2020.**
(DOCX)

**S7 Table. Emissions to water from JC oil conversion phase in Pakistan during 2019–2020.**
(DOCX)

**S8 Table. Emissions to soil from JC oil conversion phase in Pakistan during 2019–2020.**
(DOCX)

**S9 Table. Emissions to air from JC oil conversion phase in Pakistan during 2019–2020.**
(DOCX)

**S1 Graphical abstract.**
(PNG)

## Author Contributions

**Conceptualization:** Taslima Khanam, Faisal Khalid, Rana Hadi.

**Data curation:** Faisal Khalid, Wajiha Manzoor, Faizan Ullah.

**Formal analysis:** Taslima Khanam, Faisal Khalid, Andleeb Akhtar.

**Investigation:** Fariha Rehman.

**Methodology:** Faizan Ullah, Fariha Rehman.

**Resources:** Faizan Ullah, N. B. Karthik Babu.

**Software:** Ahmad Rashedi, Andleeb Akhtar.

**Supervision:** Majid Hussain.

**Validation:** Wajiha Manzoor, Ahmad Rashedi, Rana Hadi, Andleeb Akhtar, N. B. Karthik Babu, Majid Hussain.

**Visualization:** Wajiha Manzoor, Andleeb Akhtar, N. B. Karthik Babu.

**Writing – original draft:** Taslima Khanam, Faisal Khalid, Majid Hussain.

**Writing – review & editing:** Ahmad Rashedi, Rana Hadi, Faizan Ullah, Fariha Rehman, N. B. Karthik Babu, Majid Hussain.

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
