## [Decision Letter · Decision Letter 0]

10 Aug 2021

PONE-D-21-22104

Environmental sustainability assessment of biodiesel production from Jatropha curcas L. seeds oil in Pakistan

PLOS ONE

Dear Dr. Majid Hussain

Thank you for submitting your manuscript to PLOS ONE. After careful consideration, we feel that it has merit but does not fully meet PLOS ONE’s publication criteria as it currently stands. Therefore, we invite you to submit a revised version of the manuscript that addresses the points raised during the review process.

We look forward to receiving your revised manuscript.

Kind regards,

Muhammad Aamer Mehmood, Ph.D.

Academic Editor

PLOS ONE

Journal Requirements:

Additional Editor Comments (if provided):

Language of the MS needs improvement

Reviewers' comments:

Reviewer's Responses to Questions

**Comments to the Author**

1. Is the manuscript technically sound, and do the data support the conclusions?

Reviewer #1: Yes

Reviewer #2: Yes

2. Has the statistical analysis been performed appropriately and rigorously? 

Reviewer #1: Yes

Reviewer #2: Yes

3. Have the authors made all data underlying the findings in their manuscript fully available?

Reviewer #1: Yes

Reviewer #2: Yes

4. Is the manuscript presented in an intelligible fashion and written in standard English?

Reviewer #1: Yes

Reviewer #2: Yes

5. Review Comments to the Author

Reviewer #1: The manuscript “Environmental sustainability assessment of biodiesel production from Jatropha curcas L. seeds oil in Pakistan” provide the insight on the environmental impacts of the commercial scale JC biodiesel production. The paper is interesting and provide valuable information. This paper is “recommended for the publication” in the PLOS journal. However, few modifications are recommended prior to publication

• Careful revision of manuscript for formatting is recommended.

• It is recommended to provide the status of JC biodiesel production in Pakistan, as it would be useful for readers to understand the importance of this study from Pakistani perspective.

• In future assumption section, point c mentioned that soybean oil process data was consider for the analysis instead of JC oil data due to software limitations. In my opinion it would off set the parameters. Please justify this point.

• Please refer to page 7, line 175, the mentioned reference is not according to the recommended style.

• Please clarify the statement at page 9 from line 231-235, as in present form it is ambiguous.

• In manuscript it is recommended that 20% reduction in baseline fossil fuel, synthetic fertilizers, and electricity use provide significant results. It would be valuable if the impact of this condition on the JC biodiesel production is also mentioned.

• It is recommended to delete table 5 as this information was also provided in the text and this table does not provide any other significant information.

Reviewer #2: The MS submitted by Khanam et al., is focused on sustainability assessment of biodiesel production from Jatropha curcas seed oil in Pakistan using LCA.

1. Jatropha can be cultivated only on part of Pakistan, where country is heavily populated, how it can fulfill the requirement for biodiesel? Because previously Jatropha projects did not go successful in the neighbor country (India) of Pakistan due to lack of wasteland and higher cultivation costs

2. Are there any industries in Pakistan where Jatropha can be supplied as a feedstock of biodiesel?

3. Add some future directions/recommendations at the end of the abstract, purely based the findings of on your work

4. LCA of JC has been extensively studied, there

https://link.springer.com/article/10.1007/s10098-018-1558-7

Sustainability 2018, 10, 1451; doi:10.3390/su10051451

https://onlinelibrary.wiley.com/doi/abs/10.1111/j.1757-1707.2009.01014.x

https://www.hindawi.com/journals/bmri/2012/623070/

https://www.hindawi.com/journals/bmri/2012/623070/

https://www.sciencedirect.com/science/article/pii/S096085241200185X

your findings are mostly similar to these previously published LCA-based studies on JC, what is the significance novelty of your work?

6. PLOS authors have the option to publish the peer review history of their article (what does this mean?). If published, this will include your full peer review and any attached files.

Reviewer #1: No

Reviewer #2: No

---

## [Author Response · Author response to Decision Letter 0]

21 Aug 2021

RESPONSE TO REVIEWERS

REVIEWER-1 COMMENTS 

The manuscript “Environmental sustainability assessment of biodiesel production from Jatropha curcas L. seeds oil in Pakistan” provide the insight on the environmental impacts of the commercial scale JC biodiesel production. The paper is interesting and provide valuable information. This paper is “recommended for the publication” in the PLOS journal. However, few modifications are recommended prior to publication. 

Comment # 1: Careful revision of manuscript for formatting is recommended.

Answer: The manuscript is revised very carefully as per author guidelines of the PLoS ONE journal. 

Comment # 2: It is recommended to provide the status of JC biodiesel production in Pakistan, as it would be useful for readers to understand the importance of this study from Pakistani perspective.

Answer: The status of biodiesel production at prototype scale is explained in the revised manuscript in the Introduction section page no 4 as per direction of the respected Reviewer. 

Comment # 3: In future assumption section, point c mention that soybean oil process data was consider for the analysis instead of JC oil data due to software limitations. In my opinion it would offset the parameters. Please justify this point. 

Answer: As JC crude oil extraction process was not present in SimaPro software databases such as Ecoinvent v2, and soyabean and JC plant cultivation and extraction process is almost similar therefore we assumed soybean crude oil extraction process from SimaPro database for our analysis. Based on its similar process of extraction, it would have less effect on the overall results or parameters of this study. 

Comment # 4: Please to page 7, line 175, the mentioned reference is not according to the recommended style. 

Answer: The mentioned reference is changed to the recommended style of PLoS ONE as per direction of the Respected Reviewer. 

Comment # 5: Please clarify the statement at page 9 from line 231-235, as in present form it is ambiguous. 

Answer: The statement at page 9 from line 231-235 is rephrased and ambiguity is removed in the revised manuscript as per direction of the Respected Reviewer. 

Comment # 6: In manuscript it is recommended that 20% reduction in baseline fossil fuel, synthetic fertilizers, and electricity use provide significant results. It would be valuable if the impact of this condition on the JC biodiesel production is also mentioned.

Answer: Sensitivity analysis is always conducted for hotspot sources or activities, and fossil fuel, synthetic fertilizers and electricity use were the hotspot sources in the JC biodiesel production chain, therefore we performed sensitivity or scenario analysis for these hotspots’ sources, obviously reduction in these sources will lead to overall reduction in JC biodiesel production chain in Pakistan. 

Comment # 7: It is recommended to delete table 5 as this information was also provided in the text and this table does not provide any other significant information.

Answer: Table 5 is deleted from the revised manuscript as per direction of the Respected Reviewer.

REVIEWER-2 COMMENTS

Reviewer #2: The MS submitted by Khanam et al., is focused on sustainability assessment of biodiesel production from Jatropha curcas seed oil in Pakistan using LCA.

Comment # 1. Jatropha can be cultivated only on part of Pakistan, where country is heavily populated, how it can fulfill the requirement for biodiesel? Because previously Jatropha projects did not go successful in the neighbour country (India) of Pakistan due to lack of wasteland and higher cultivation costs.

Answer: Site suitability analysis was also performed for JC bioenergy plantation in Khyber Pakhtunkhwa province of Pakistan based on water footprint and yield of JC seeds production, which showed that southern part of the country is more suitable for JC bioenergy plantation as compared to the northern part. Southern part is arid and semi-arid with less population and more wasteland. As far as the neighbor country India is concerned regarding JC cultivation, the population of India is more than a billion whereas Pakistan population is less than India and then population in KP is less than population in Punjab province of Pakistan. The cultivation cost or economic analysis was not part of this study, it will be investigated in near future in my MS student upcoming research project. In addition, there is huge JC cultivation going on in KP province of Pakistan supported by Government, National and International agencies in Pakistan. 

2. Are there any industries in Pakistan where Jatropha can be supplied as a feedstock of biodiesel?

Answer: Although JC biodiesel is produced as prototype in different Labs of various Universities and organizations, however the first biodiesel Plant was installed in Bahawalpur on trial basis and after success of this plant there will be many more in Pakistan where JC seeds will be supplied as feedstock of biodiesel. Moreover, Pakistan desires to blend 10% biodiesel in fossil diesel since 2025, therefore huge research has been going on biodiesel production from various biomass feedstocks in Pakistan. 

3. Add some future directions/recommendations at the end of the abstract, purely based the findings of on your work.

Answer: Some future directions/recommendations are added at the end of the abstract of this study based on outcomes of this study as directed by Respected Reviewer-2. 

4. LCA of JC has been extensively studied, there

https://link.springer.com/article/10.1007/s10098-018-1558-7

Sustainability 2018, 10, 1451; doi:10.3390/su10051451

https://onlinelibrary.wiley.com/doi/abs/10.1111/j.1757-1707.2009.01014.x

https://www.hindawi.com/journals/bmri/2012/623070/

https://www.hindawi.com/journals/bmri/2012/623070/

https://www.sciencedirect.com/science/article/pii/S096085241200185X

your findings are mostly similar to these previously published LCA-based studies on JC, what is the significance novelty of your work?

 Answer: LCA of JC has been extensively studied in other countries of the World, however, this is the ever first study on LCA of JC biodiesel in Pakistan. The novelty of this study is variation in the system boundary of the present study with rest of the global studies conducted and variation in the inputs of the JC biodiesel production chain in Pakistan for example Kohloo method (A Kohloo is a machine which is used for the extraction of oil from JC seeds manually or mechanically, the Kohloo machine was run by bull in the present study. These variation in JC biodiesel production chain of Pakistan with rest of the global countries make this study very innovative and novel and provide valuable insights for decision making authority regarding bioenergy promotion in Pakistan. Second, according to IPCC recommendation, an environmental profile analysis is mandatory for any prototype before its large-scale commercialization, therefore, this is the ever first LCA of biodiesel prototype in Pakistan. 

Vote of Thanks!

We the authors of this paper are extremely thankful to the Respected Reviewers and Editor-in-Chief PLoS ONE for considering our research work in this prestigious journal. The Respected Reviewers comments/suggestions were highly valuable and incorporation of all the comments improved the quality of the revised manuscript. We hope that PLoS ONE will consider our future research work too. Thank you very much. 

Best Regards,

Dr. Majid Hussain

Assistant Professor/Corresponding author 

Department of Forestry and Wildlife Management 

University of Haripur, Pakistan

---

## [Editor Report · Decision Letter 1]

27 Sep 2021

Environmental sustainability assessment of biodiesel production from Jatropha curcas L. seeds oil in Pakistan

PONE-D-21-22104R1

Dear Dr. Majid Hussain,

We’re pleased to inform you that your manuscript has been judged scientifically suitable for publication and will be formally accepted for publication once it meets all outstanding technical requirements.

Kind regards,

Muhammad Aamer Mehmood, Ph.D.

Academic Editor

PLOS ONE

Additional Editor Comments (optional):

Please remove the column "Compartment" from the Tables provided as Supplementary material. The compartment information can be provided in the Table Caption, once. 
---

## [Editor Report · Acceptance letter]

29 Oct 2021

PONE-D-21-22104R1 

Environmental sustainability assessment of biodiesel production from *Jatropha curcas* L. seeds oil in Pakistan 

Dear Dr. Hussain:

I'm pleased to inform you that your manuscript has been deemed suitable for publication in PLOS ONE. Congratulations! Your manuscript is now with our production department. 

Kind regards, 

on behalf of

Dr. Muhammad Aamer Mehmood 

Academic Editor

PLOS ONE